# Unsupervised Detection of Multiple Sleep Stages Using a Single FMCW Radar

Young-Keun Yoo ⬤, Chae-Won Jung and Hyun-Chool Shin *

Department of Electronic Engineering, University of Soongsil, Seoul 06978, Republic of Korea
* Correspondence: shinhc@ssu.ac.kr; Tel.: +82-828-7165

**Abstract:** The paper proposes a unsupervised method for detecting the three stages of sleep—wake, rapid eye movement (*REM*) sleep, and non-*REM* sleep—using biosignals obtained from a 61 GHz single frequency modulated continuous wave (FMCW) radar. To detect the subject's sleep stages based on non-learning techniques, the breathing and movement information characteristic of each sleep stage was extracted from the radar signals of the subject acquired in the sleep state and used as the feature factor tailored to the research objective. The experimental results derived from the clinical data obtained in the actual polysomnography (PSG) environment using FMCW radar show an average of 68% similarity to the actual three sleep stages observed in PSG. These results indicate the feasibility of using the FMCW radar sensor as an alternative to the conventional PSG-based method that poses multiple limitations to sleep-stage detection.

**Keywords:** sleep stage detection; contactless; non-learning; vital detection; signal processing; FMCW radar

## 1. Introduction

Sleep is an essential physiological activity for survival that restores the physical abilities of the human body and brain, conserves energy, secretes hormones, and stores memories, among other functions [1]. Decreased sleep quality due to sleep deprivation can lead to various mental and physical difficulties, including poor immunity, elevated risks of chronic diseases, weight gain, increased depression, decreased libido, impaired emotional control, and worse memory performance [2]. Therefore, an accurate analysis of sleep quality through sleep-stage detection is important for diagnosing health conditions [3].

Polysomnography (PSG) is the most widely used standard for monitoring sleep stages and evaluating sleep quality. PSG is a comprehensive test that passively diagnoses a subject's sleep quality by recording complex physiological indicators, including brain waves, blood oxygen levels, breathing and heart rates, and eye and leg moments during sleep [4]. Although PSG is a highly accurate and reliable method for assessing sleep stages based on brain wave changes during sleep [5], it has many limitations. First, PSG is unsuitable for long-term continuous monitoring because the test subjects tend to find it uncomfortable to have at least 22 sensing electrodes required for the test attached to the body [6]. In addition, PSG is cost-ineffective, as the passive test requires reading by trained sleep technologists in laboratories equipped with specialized testing equipment [7,8]. In addition, one-time measurements force the results to rely on the physical condition of the subject on the test day [9]. These shortcomings cause discomfort to the test subject, and as a result, fail to accurately reflect sleep quality. Therefore, there is a need for research that enables accurate detection of sleep stages while resolving PSG issues.

To overcome the challenges of PSG, many techniques have been adopted to automatically monitor sleep stages under conditions that are as similar as possible to actual sleep. A wrist actigraph, which detects a subject's wakefulness by measuring the amount of physical activity based on the acceleration of the application area, is the most commonly used device

in sleep-related clinical studies. When the device cannot be worn on the wrist, it can be worn on the torso or ankle [10]. In addition, this technique allows an easy distinction between the wake and sleep states of a target based on the acceleration generated by its movement [11]. However, this technique has low reliability. Errors due to the subject's wearing conditions and hand dominance influence the sleep stage categorization results, making it difficult to identify detailed sleep stages, such as the rapid eye movement (*REM*) sleep stage [12].

Moreover, distinguishing the *REM* sleep stage using motion sensors that detect the awake state based on the subject's body movements is difficult because additional biometric information, such as breathing and heart rate, is not easily acquired [13,14]. Unlike actigraph devices and motion sensors that detect only the body part(s) equipped with sensors, pressure sensors, which can measure the subject's overall movement and provide additional biometric information, such as breathing and heart rate, can be used for continuous data collection in the form of an ordinary bed. However, pressure sensors are expensive, and sleep-stage results vary depending on the subject's weight [15,16]. In addition, acoustic sensors, which identify the sleep stage by collecting sound information caused by the subject's breathing and snoring during the test, have low measurement accuracy because of their inability to utilize the subject's motion information and the influence of noise generated by the surroundings [17].

Contactless sleep-stage detection techniques based on various radar sensors have been proposed to monitor sleep state in a more natural environment by resolving the limitations of the contact measurements involved in PSG and addressing the shortcomings of the methods introduced in previous studies. In existing studies on radar-based sleep-stage detection, various learning methods with biometric features extracted using CW radar sensors have been used to divide the sleep state into three (wake, *REM*, and NREM) [18,19] or four stages (wake, *REM*, light sleep, and deep sleep) [20,21]. In addition, sleep-stage-detection studies have been conducted using biometric information extracted from additional equipment, such as acoustic sensors and radar sensors, as machine learning feature factors to improve detection accuracy [22]. In addition, some studies have used UWB radar sensors to classify sleep states into three stages [23,24]. However, these methods have limitations because of the significant amount of signal processing computation required owing to the nature of sleep tests that measure data overnight.

Moreover, some studies have used two or nine radar sensors to identify sleep stages, using machine learning, where detection and classification accuracy improve as the number of feature factors increases [25,26]. In contrast, some previous studies performed non-learning detection using only biometric information obtained with IR-UWB radar to overcome the signal processing limitations of learning-based, radar-detected sleep-stage detection. However, only two sleep stages can be distinguished [27].

No studies have employed frequency modulated continuous wave (FMCW) radar to detect sleep stages using nonlearning techniques. FMCW radar is easy to popularize because it is less costly than pulse radar and is suitable for breathing and heart-rate detection according to the microdisplacement of the target, as it has a wide bandwidth for radio transmission [28]. Compared with the CW radar, which transmits a single frequency signal and cannot distinguish the target from interfering objects, the FMCW radar obtains Doppler and range information about the detection target through frequency modulation of the transmitted and received signals, making it advantageous for detecting biosignals [29].

Therefore, this study introduces a method for the non-learning detection of the three sleep stages using a single FMCW radar with an intermediate frequency of 61 GHz to overcome the limitations of previous studies on PSG and sleep-stage detection. The *REM* sleep stage was primarily detected using FMCW radar-detected respiration information. In addition, the subject's in-sleep movement was estimated and reflected in the results of the *REM* sleep-stage detection to mitigate false *REM* identification caused by *REM*-sleep behavior disorders. In this modification, the interval during which the subject's body movement was too large was considered as the wake state. Owing to the nonideal nature

of clinical data, this step poses various challenges in determining the threshold value based on biometric information. Therefore, the threshold was set experimentally based on the acquired data instead of an arbitrary value to address these issues. Nonlearning-based three-stage sleep detection was performed using the estimated thresholds and biometric information. The accuracy of the proposed method was verified by comparing it with sleep-stage results from PSG readings.

## 2. FMCW Radar Signal Processing

Radar is a contactless sensing device that detects a target by estimating its orientation, distance, and speed information using the received signal—a radio wave transmitted to the target and returned to the device after being reflected off the target's surface. Radars are classified into four types based on the hardware signal transmission method: pulse, CW, SFCW, and FMCW. Furthermore, based on the signal processing method, they are classified into two types: Doppler and non-Doppler radars.

The FMCW radar used here was a type of doppler sensor that transmits and receives electromagnetic waves whose frequency increases or decreases according to time. By analyzing the frequency difference between the transmitted signal of the FMCW radar and the received signal reflected from the target at a certain distance, the time delay and phase change of the signal can be detected. Using the variability of the signal components according to the movement of the target, both range and speed information of the target can be extracted. Radar transmits a chirp signal from its signal generator, the frequency of which increases linearly with time towards an object. The signal reflected from the object and received through the receiving antenna is then combined with the transmitted signal using a mixer. After processing the integrated signal through a low-pass filter, the frequency difference between the transmitted and received signals was obtained and converted into a digital signal using an analog-to-digital converter for signal processing.

The transmitted signal $T_x(t)$ is expressed by (1). Equation (2) includes the time delay due to the additional distance traveled by the received signal $R_x(t)$ reflected from the target object.

$$T_x(t) = M_r(t) \cdot cos(2\pi(f_{carrier} + \frac{BW \cdot t}{2 \cdot T_c}) \cdot t) \tag{1}$$

$$R_x(t) = M_R(t) \cdot cos(2\pi(f_{carrier} + \frac{BW \cdot (t - t_d)}{2 \cdot T_c}) \cdot (t - t_d)) \tag{2}$$

$M(t)$ is the magnitude of the transmitted and received signals, $f_{carrier}$ is the carrier frequency, $T_c$ is the duration of the chirp signal, and $t_d$ is the time delay required for the received signal to be reflected and returned. A frequency difference is observed between the transmitted and received signals owing to the target object distance; this difference is defined as an intermediate frequency. As shown in (3), the signal $x(t,n)$ containing the intermediate frequency component was extracted by multiplying the transmitted and received signals using the mixer, a circuit element within the radar, and applying a low-frequency filter [30].

$$x(t,n) = \sum_r M(t,r) \cdot cos(2\pi \cdot f_r \cdot n \cdot + P(t,r)) \tag{3}$$

where $n$ refers to the sampling duration, $r$ represents the distance information, $f_r$ stands the intermediate frequency, and $P(t,r)$ refers to the phase component of distance with time. A fast Fourier transform was used to extract the magnitude, frequency, and phase components of the intermediate-frequency signal, as follows:

$$X(t,r) = \sum_{n=1}^{N} x(t,n) \cdot exp^{(-\frac{j \cdot 2\pi \cdot k \cdot n}{N})} \tag{4}$$

where $N$ denotes the number of samples per chirp. When an object is placed at the distance $r_m$ from the radar, the magnitude and phase of the signal are defined using $X(t, r)$ as follows:

$$M(t, r_m) = 2|X(t, r_m)| = \frac{M_0}{4\pi \cdot (r_m)^2} \cdot r_m \tag{5}$$

$$P(t, r_m) = \angle X(t, r_m) = \frac{4\pi \cdot f_c}{c} \cdot r_m \tag{6}$$

Here, $M_0$ refers to the magnitude of the transmitted signal and $c$ represents the speed of light. The magnitude and phase of the intermediate frequency obtained using the radar contain the information on distance to the target. Therefore, biometric information such as respiration, heart rate, and movement can be extracted by utilizing changes in the components of the intermediate-frequency signal that occur because of the micro-displacement of the target.

## 3. Sleep Stage Detection

### 3.1. Physiological Characteristics of the Sleep Stages

This study performed three-stage sleep detection using physiological characteristics specific to each sleep stage. This method differs from PSG, which monitors sleep stages based on the variability in the subject's EEG measurements during sleep [31].

First, the entire wake–sleep cycle can be broadly divided into wake and sleep states. The sleep state is further divided into *REM* and Non-*REM* sleep stages. In the wake state, the subject's body movements were observed more frequently than in other stages; breathing rate and heart rate were higher than those in the sleep state, and their cycles were irregular. In the *REM* sleep stage, breathing and heart rate were higher, with irregular changes over time, compared with the intervals adjacent to the non-*REM* sleep stage. Normal *REM* sleep rarely involves body motion, but patients with *REM*-sleep behavior disorders experience sudden body movements during *REM* sleep. Lastly, the non-REM sleep stage, a very stable phase in the sleep state, is characterized by little body movement and relatively low breathing and heart rates compared to the wake and *REM* sleep stages.

### 3.2. Bio-Signal Detection

Figure 1 shows the subject's breathing data collected using the intermediate-frequency signal of the FMCW radar. Figure 1a shows the change in the intermediate-frequency signal over time, where the regular changes in amplitude are caused by the body's micromovements resulting from the subject's respiratory activity. The larger the amplitude changes of the breathing signal, which can be observed in the color bar on the right, the greater the strength of the received intermediate-frequency signal. Here, a change in the radar signal is defined as respiration. The subject's respiratory signal was specified using a phase component corresponding to a certain distance (distance between the radar and the detection target) for the entire signal detection range, by using the FMCW radar to minimize the signal interference due to clutter in the surrounding environment. The respiratory signal was extracted using the distance determined based on the coherence between the magnitude and phase of the intermediate-frequency signal [32]. As shown in Figure 1b, the waveform changed over time, and the peak points of the radar-detected breathing signals used in this study were consistent with the abdominal pressure-sensor-measured signals used as the reference ground truth. Thus, the potential application of the respiratory information extracted using the FMCW radar was verified.

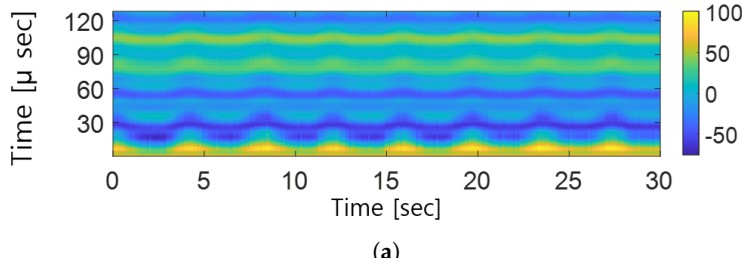

(a)

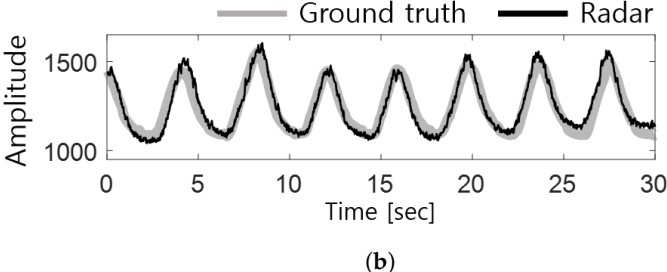

(b)

**Figure 1.** Comparison of breathing signals. (**a**) Intermediate frequency signals of the radar. (**b**) Comparison of respiratory signals acquired with the radar and breathing sensors.

### 3.3. Detection of REM Sleep Stage

An automatic sleep-stage detection method, in which sleep stages are manually identified, was proposed to address the inconvenience associated with PSG. First, utilizing the characteristics of the *REM* stage and increased respiration rate compared to the non-*REM* sleep stage, the subject's *REM* sleep stage, $REM(t)$, is primarily identified as follows:

$$REM(t) = \begin{cases} 1, & \text{if } RR(t) > RR_{thres} \\ 0, & \text{otherwise.} \end{cases} \tag{7}$$

where $RR(t)$ is the respiratory rate measured per minute and is estimated from the radar signals acquired during sleep. The respiratory rate was calculated using a zero-crossing algorithm that detected the point at which the amplitude of the breathing signal passed zero at the radar-based characteristic distance. $RR_{thres}$ represents the subject's ordinary respiratory rate assigned to distinguish the *REM* sleep stage and was set as follows using the mean and variance of the breathing rate for the entire sleep duration.

$$RR_{thres} = \frac{1}{T} \cdot \sum_{t=1}^{T} RR(t)$$
$$+ \sqrt{\frac{1}{T-1} \cdot \sum_{t=1}^{T} \left| RR(t) - \frac{1}{T} \cdot \sum_{t=1}^{T} RR(t) \right|^2} \tag{8}$$

The results of using the *REM* sleep-stage-detection algorithms suggested by (7) are shown in Figure 2. The intervals with a breathing rate above the threshold were marked as the *REM* sleep section, as shown in Figure 2b, and the resulting graph is shown in Figure 2c. In Figure 2c, the interval where the PSG reading results shown in Figure 2a differ from the radar-detected *REM* sleep-stage results is indicated in blue. This interval is where false detections occur because of the increased respiration rate caused by movement. Thus, the subject's body movement during sleep needs to be detected and utilized as an additional feature factor to improve the accuracy of *REM* sleep-stage detection.

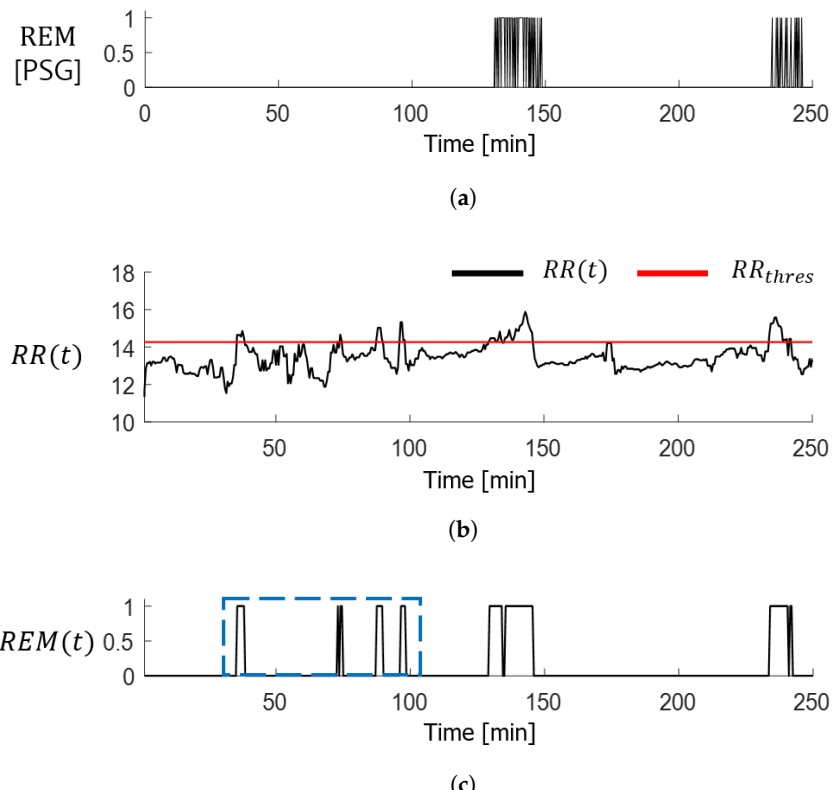

**Figure 2.** *REM* sleep stage detected with the FMCW radar. (**a**) PSG results. (**b**) Changes in breathing rate per minute (**c**) Radar-detected *REM* sleep stage.

### 3.4. FMCW Radar-Based Movement Detection

The subject's motion $Mov(t)$ is defined by accumulating the differences in the radar signal magnitude over time to measure the body movement observed during the *REM* sleep stage [33].

$$Mov(t) = \sum_{t=1}^{T} |m(t) - m(t-1)| \qquad (9)$$

where *T* is the time window for the movement extraction. Figure 3 shows the change in the movement quantification indicator $Mov(t)$ according to body movement. Figure 3a shows that a distortion in the intermediate-frequency signal occurred over the interval in which the subject moved in the sleep state. In the same interval, the subject's movement, as detected by the radar, increased sharply, as shown in Figure 3b.

To investigate the application potential of radar-based motion estimation indicators, the sudden movement intervals were compared with the intervals in which the numerical changes in acceleration for each body part were measured using an accelerometer. Among the 14 Perception Neuron Studio motion sensors (Noitom Inc., Miami, FL, United States) attached to the body, and accelerometer data obtained from the three body parts where the actual object movement occurred were used as the ground truth for comparison. As the target movement detected by the FMCW radar coincided with the intervals of the numerical changes in body movement measured using an accelerometer, it was found that $Mov(t)$ reflects the subject's body movement.

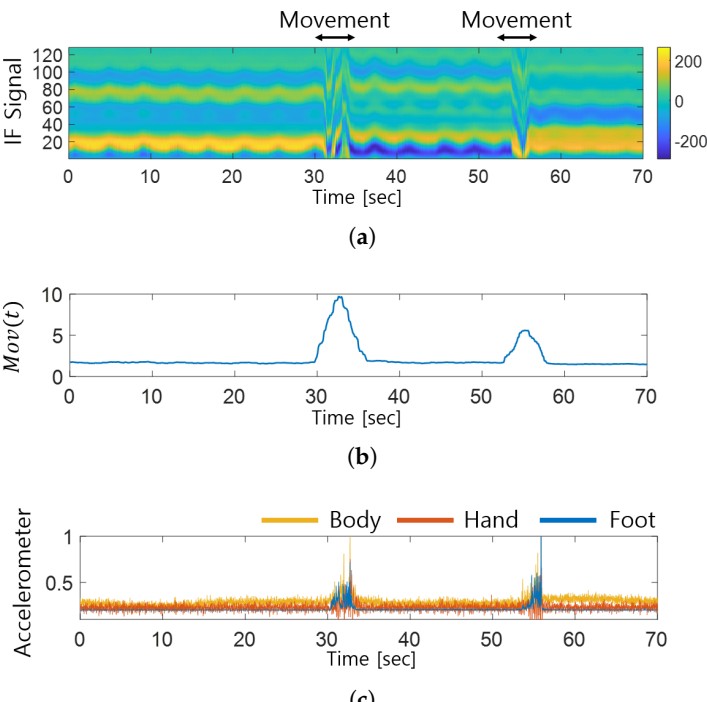

**Figure 3.** FMCW radar-based movement-detection results. (**a**) Intermediate frequency signal of the radar. (**b**) Motion index $Mov(t)$. (**c**) Accelerometer data for different body parts for verification.

*3.5. Reducing the False Detection of REM Sleep Stage*

The updated *REM* sleep stage, $REM'(t)$, was determined as below by additionally considering the subject's motion data and the *REM* sleep stage detected primarily using the breathing-rate changes. Changes in $Mov(t)$ due to the subject's momentary movements were minimized by applying a median filter on the radar-detected motion values. When the subject's body movement increases above the threshold value $Mov_{thres}$ in the primarily identified *REM* sleep interval, the motion is assessed as a sudden movement due to an *REM*-sleep behavior disorder or a single-episode activity during sleep. This motion was excluded from the primary detection of *REM* sleep.

$$REM'(t) = \begin{cases} 0, & \text{if } REM(t) = 1 \\ & \& \ Mov(t) > Mov_{thres} \\ REM(t), & \text{otherwise.} \end{cases} \tag{10}$$

Here, $Mov_{thres}$, which is defined as the sum of the mean and variance of the subject's movements over the total data collection duration, represents the subject's average body movements during sleep.

$$Mov_{thres} = \frac{1}{T} \cdot \sum_{t=1}^{T} Mov(t)$$
$$+ \sqrt{\frac{1}{T-1} \cdot \sum_{t=1}^{T} \left| Mov(t) - \frac{1}{T} \cdot \sum_{t=1}^{T} Mov(t) \right|^2} \tag{11}$$

Figure 4 shows examples of *REM* sleep-stage detection using (10) for the four subjects. The detection results in Figure 4a, which are from normal subjects who did not experience body movements during *REM* sleep, agree with the PSG reading and radar sensing results. A comparison of the detection results $REM(t)$ and $REM'(t)$ according to the subject's motion during *REM* sleep in Figure 4b–d, shows that the false *REM* stage detection interval,

indicated in blue, improved and matched the PSG results after reflecting the subject's body motion data.

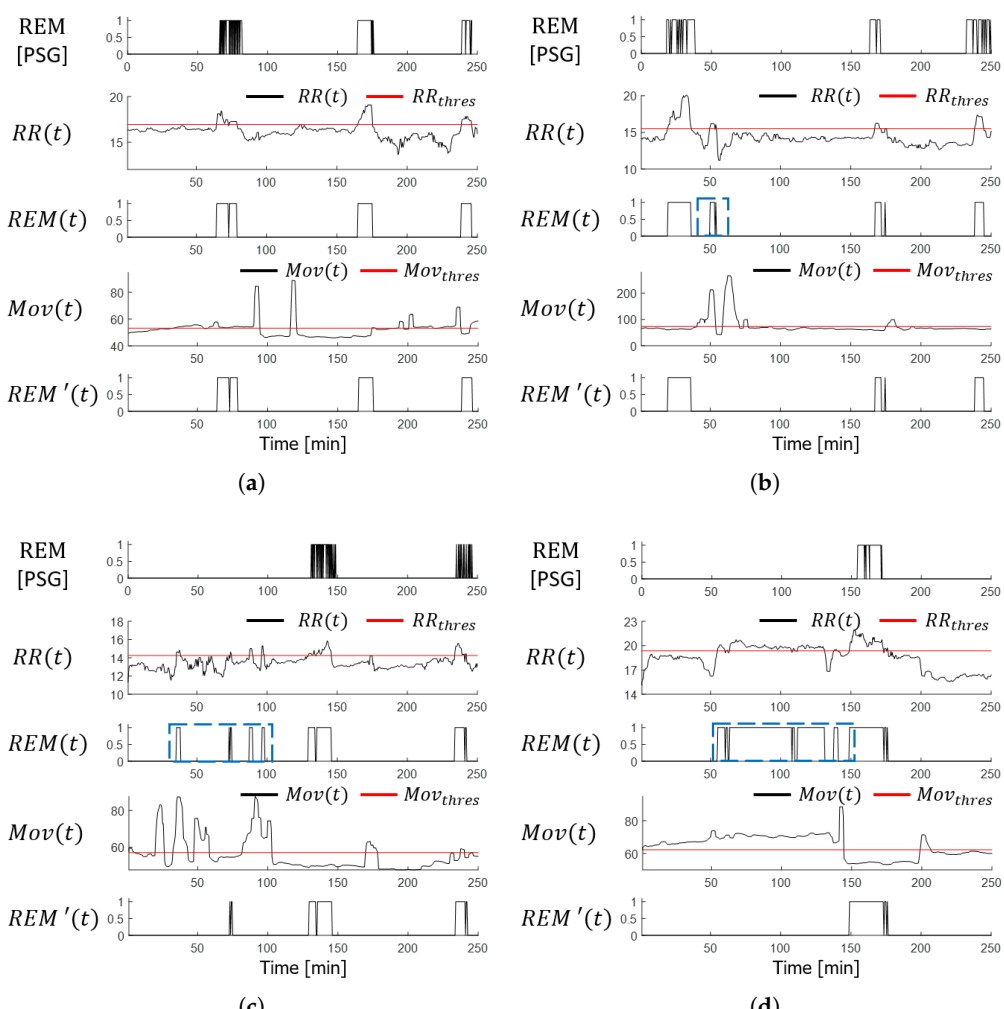

**Figure 4.** FMCW radar-based REM sleep-stage-detection results. (**a**) Case of normal detection. (**b**–**d**) Case with false detection.

### 3.6. Wake State Detection

The wake interval $W(t)$ was determined, as shown in (12), by reflecting the characteristics of the wake state, during which the subject's body motion became more active compared to the sleep state.

$$W(t) = \begin{cases} 1, & \text{if } Mov(t) > \frac{1}{T} \cdot \sum_{t=1}^{T} Mov(t) \\ 0, & \text{otherwise.} \end{cases} \tag{12}$$

The wake interval represents the scenario in which the subject's movement, detected using the sensing radar, increases above the average of the motion estimation indicator for the entire duration. The detected wake intervals were compared with the PSG readings. Figure 5 shows examples of wake-stage detection obtained by applying (10) to the radar signals acquired from the four subjects. These results are in agreement with the PSG results.

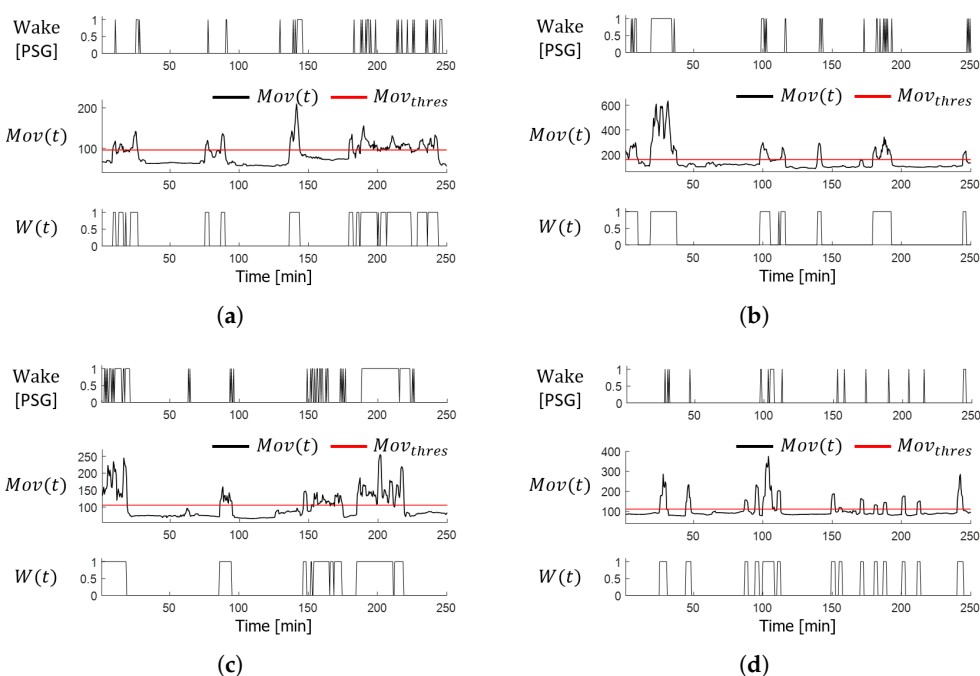

**Figure 5.** FMCW radar-based wake stage detection results. (**a**) Subject 1. (**b**) Subject 2. (**c**) Subject 3. (**d**) Subject 4.

## 4. Experimental Results

### 4.1. Experimental Setup

Figure 6a shows the experimental setup in which the clinical sleep data were obtained from the PSG test subjects. Clinical sleep data were collected from 85 participants in the Polysomnographic Laboratory at Pusan National University Yangsan Hospital. The radar was installed 0.4 m above the headboard facing the subject's chest and used for an average of more than 6 h simultaneously with PSG. The PSG readings provided by clinical specialists were used as the reference ground truth and were compared with the results obtained from the proposed sleep-stage detection algorithm. A single-channel FMCW radar (BitSensing Inc., Seoul, Korea) [34], as shown in Figure 6b, was used to collect and record the biometric information of the PSG subjects. The radar specifications are listed in Table 1. This radar, a 3-channel radar comprising one transmitting antenna and three receiving antennas, which transmits millimeter waves with an intermediate frequency of 60 GHz, is an indoor-sensing radar capable of detecting the distance and Doppler information of a target object located within the range of 0–0.75 m before the radar. We used only the data received through the single channel with the best SNR.

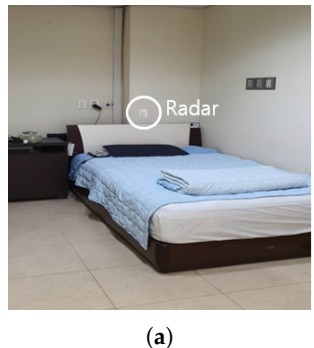

(**a**)

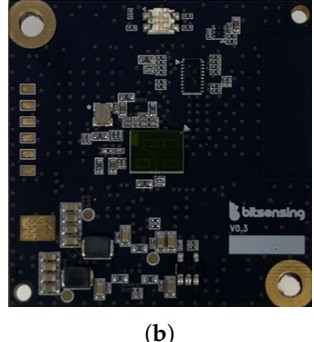

(**b**)

**Figure 6.** Experimental setup. (**a**) Clinical data acquisition environment. (**b**) Single-channel FMCW radar.

**Table 1.** FMCW radar specifications.

| Parameters | Value |
| --- | --- |
| Center Frequency | 60 GHz |
| Chirp duration | 300 μs |
| Sampling Frequency | 1 MHz |
| Scan interval | 100 ms |
| Bandwidth | 3.75 GHz |
| Number of Tx antenna | 1 |
| Number of Rx antenna | 3 |

*4.2. Sleep-Stage Detection Results*

Three sleep stages (wake, *REM*, and non-*REM*) were detected using the breathing and movement data of the subjects extracted using the FMCW radar. Figure 7 presents the results of the sleep-stage detection performed on the two subjects as examples. The similarity between the radar-based sleep-stage detection results and the PSG results for the entire test duration was confirmed.

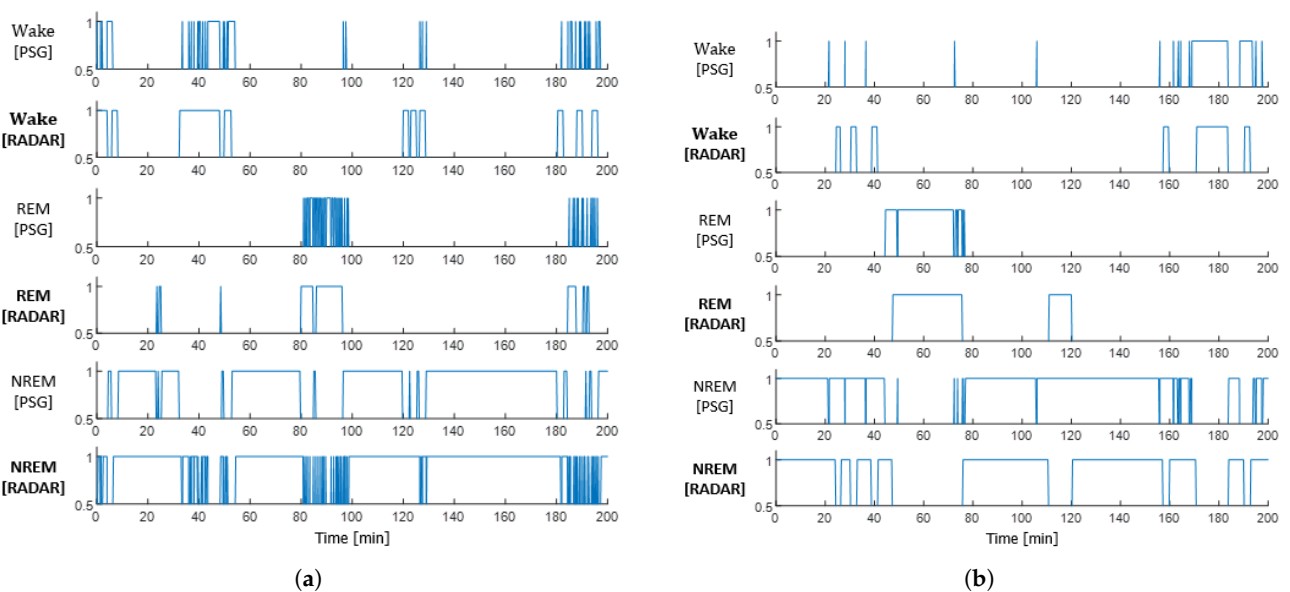

(**a**)            (**b**)

**Figure 7.** Three-stage sleep-detection results using FMCW radar. (**a**) Subject 1. (**b**) Subject 2.

In addition, a confusion matrix was employed to assess the accuracy of FMCW radar-based three-stage sleep detection. Figure 8 shows the results of integrating and labeling PSG readings and three radar-detected sleep stages at 30 s (1-epoch) intervals for stage-specific accuracy evaluation. The wake stage was matched to 3, *REM* stage to 2, and non-*REM* stage to 1.

*4.3. Accuracy Analysis*

As determining the stage-specific risks associated with radar-based sleep-stage detection results was not the research objective, accuracy was evaluated using the ratio of correctly identified sleep stages to all cases. Table 2 lists the average accuracy of the proposed sleep-stage detection algorithm tested on 85 subjects. The detection accuracies for the awake, *REM* sleep, and non-*REM* sleep stages were 64.37%, 83.51%, and 58.98%, respectively. The average detection accuracy for all three stages was 68.91%.

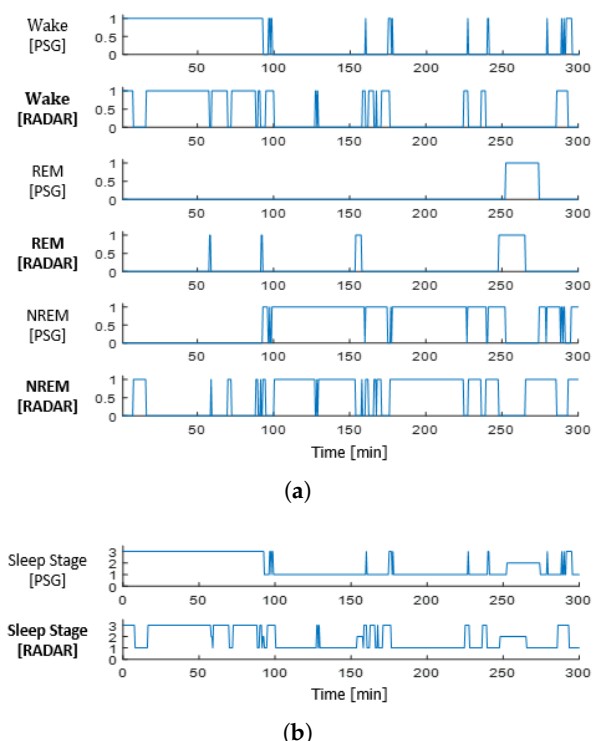

(**a**)

(**b**)

**Figure 8.** Labeling in sleep-stage-detection results. (**a**) Sleep-stage-detection results before labeling. (**b**) Sleep-stage-detection results after labeling.

**Table 2.** Sleep-stage-detection accuracy.

| Algorithm | Radar in Application | Evaluation Criteria | Number of Stage | Use of Learning-Based Method | Detection Accuracy [%] |
|---|---|---|---|---|---|
| [18] | CW Doppler | Accuracy | 4 | O | W/R/L/D 81.8/85.5/78.6/81.2 |
| [23] | IR UWB | Accuracy | 4 | O | Four-stage Average 82.6 |
| [20] | CW Doppler | Precision Recall F1-Score | 3 | O | W/S-(P)86.0/(R)86.5/(F)86.2 R/NR-(P)75.5/(R)75.4/(F)75.8 |
| [24] | IR UWB | Accuracy | 3 | O | Three-Stage Average 72.93 |
| [35] | Bio | Precision | 3 | O | Three-Stage Average 75.13 |
| [25] | Micro Doppler | Precision | 3 | O | Three-Stage Average 68.10 |
| [26] | Micro Doppler | Accuracy | 3 | O | Three-Stage Average 57.10 |
| [27] | IR UWB | Recall | 2 | X | Three-Stage Average 75.00 |
| Proposed | FMCW | Accuracy | 3 | X | Three-Stage Average 68.91 |

The previous research [27] used a pulse-based IR-UWB radar for sleep-stage detection. Although IR-UWB radar has merits in identifying targets behind objects, it has a limited transmission power compared to FMCW radar, and its design is very complicated and the cost greatly increases due to a high-speed analog to digital converter (ADC). In addition, although the detection accuracy in [18,20,24–26] is higher than that of the proposed method, those were required a training procedure for sleep-stage detection. Ref. [27] used an unsupervised method, but it focused on neonates and detected only sleep and wake stages without considering the *REM* stage. In the *REM* sleep stage, muscle atony increases and causes sleep breathing disorders, such as sleep apnea and hypo-apnea, so it is an important stage that needs to be detected in sleep monitoring research [36]. Although the

accuracy of the proposed method is not high compared with those of conventional ones, the proposed method is valuable, in that it detects three stages, including *REM* sleep, in an unsupervised manner.

## 5. Conclusions

This paper proposes a non-learning-based, three-stage sleep-detection algorithm using FMCW radar sensors. The subject's respiration and movement information was extracted to detect the three stages (wake, *REM*, and Non-*REM*) in the sleep state. Compared to existing research on non-learning-based, two-stage sleep detection, the *REM* sleep stage was additionally detected, and the subject's motion information was also used to reduce *REM* sleep misdetection caused by the movement of patients with *REM*-sleep behavior disorder. The performance of the proposed method was evaluated with real clinical data obtained from patients using FMCW radar. Compared with the PSG results, the three-stage detection accuracy was 88.8% for a single subject and an average of 68.91% for the 85 subjects. The results suggest that the proposed sleep-stage detection method can be utilized instead of existing PSG and sleep-stage monitoring methods. However, in the hospital, various sleep stages and states are monitored with various sensors, including brain wave sensors. Neurophysiological information that can be detected using brain waves could not be detected using radar. Therefore, a hybrid sleep monitoring system combing radar and brain waves needs to be considered for future work.

**Author Contributions:** Conceptualization, Y.-K.Y.; data curation, Y.-K.Y. and C.-W.J.; formal analysis, Y.-K.Y. and C.-W.J.; investigation, Y.-K.Y.; methodology, Y.-K.Y. and C.-W.J.; project administration, H.-C.S.; resources, H.-C.S.; software, Y.-K.Y. and C.-W.J.; supervision, H.-C.S.; validation, H.-C.S.; visualization, Y.-K.Y. and C.-W.J.; writing—original draft, Y.-K.Y.; writing—review and editing, H.-C.S. All authors have read and agreed to the published version of the manuscript.

**Funding:** This work was supported by institute of Information and communications Technology Planning and Evaluation (IITP) grant funded by the Korean government (MSIT) (No. 2021-0-00305).

**Institutional Review Board Statement:** Not applicable.

**Informed Consent Statement:** Informed consent was obtained from all subjects involved in the study.

**Data Availability Statement:** Not applicable.

**Conflicts of Interest:** The authors declare no conflict of interest.

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
