# Peer review of "Unsupervised Detection of Multiple Sleep Stages Using a Single FMCW Radar"

_applsci, doi:10.3390/app13074468_

Round 1

Reviewer 1 Report

This study proposes and evaluates a method for detecting the three stages of sleep—Wake, 1 REM, Non-REM—by applying non-learning-based methods to the bio-signal changes of the polysomnog- 2 raphy (PSG) subject in sleep state measured using a 61-GHz single frequency modulated continuous  wave (FMCW).  This work needs important improvements which are listed as follows.

1. The title of the paper must be easy to understand everyone. please revise

2. The first sentence of the abstract is not clear, and the achievements are discussed in the abstract accuratly please revise. 

3. no need define abbreviaiton in the keywords.

4.  please keep space among reference citation and text like line 16 and 20.

5.  The presentation of the paper is too weak lke "The FMCW (frequency modulated continuous wave) radar used here—a kind of 108 Doppler sensor— can detect the amplitude and time delay of the signal and the phase 109 change of the transmitted and received signals. " The explanation of FMCW is wrong and it should be defined for the first time. no in the mid text. 

6. Figures 1 to 5 are difficult to read. 

7.  Standard reference format is not followed. 

3. 

Author Response

Manuscript ID: applsci-2297904

Article Title: “Non-Learning Based Multiple Sleep Stages Detection Using Single FMCW Radar

Dear reviewer,

Thank you for valuable comments, and an opportunity to revise the paper.

We are uploading our point-by-point response to the comments (below) (response to reviewers), and an updated manuscript.

Best regards,

Hyun-Chool Shin

Reviewer 2 Report

The manuscript proposes a method that utilizes a 61-GHz single frequency modulated continuous wave (FMCW) to measure the breathing and movement information of the subject during sleep and classify sleep into three stages: Wake, REM, and Non-REM. The method overcomes some limitations of the traditional PSG method and has lower computational complexity compared to machine learning-based sleep stage classification methods. However, the detection accuracy of the proposed method is relatively low. 

Here are my comments for this paper:
1. It is recommended to provide the full name of abbreviations when they are first introduced in the paper, such as "FMCW" and "REM" in Section 1.

2. Please increase the resolution of the images in the paper.

3. In Section 4.3, should "[23]" be changed to "[27]" in the sentence "Although the sleep detection performance of the current study is lower than that of the prior non-learning-based detection study [23] by approximately 6%"?

4. The authors are suggested to add a more detailed discussion section before the conclusion to explain the superiority of the proposed method, such as elaborating on the significance of increasing the number of sleep stage classifications at the cost of accuracy in sleep stage detection studies.

5. Have other methods been attempted to determine thresholds and improve the accuracy of sleep stage detection?

6. Have the authors analyzed why there is a large difference in the detection accuracy of the three sleep stages using the proposed method?

7. Mention any shortcomings of the proposed study if any, and future works.

Author Response

(The authors gave the same response as above.)

Round 2

Reviewer 1 Report

Thanks to the authors my comments are addressed.